# Relationships among Physicochemical, Microbiological, and Parasitological Parameters, Ecotoxicity, and Biochemical Methane Potential of Pig Slurry

María Eugenia Beily [1], Brian Jonathan Young [1,*] , Patricia Alina Bres [1], Nicolás Iván Riera [1], Wenguo Wang [2], Diana Elvira Crespo [1] and Dimitrios Komilis [3]

1    Instituto Nacional de Tecnología Agropecuaria (INTA), Instituto de Microbiología y Zoología Agrícola (IMyZA), Hurlingham 1686, Argentina
2    Key Laboratory of Development and Application of Rural Renewable Energy, Biogas Institute of Ministry of Agriculture and Rural Affairs, Chengdu 610041, China
3    Department of Environmental Engineering, Democritus University of Thrace, 67100 Xanthi, Greece
*    Correspondence: young.brian@inta.gob.ar; Tel.: +54-11-4621-0799 or +54-11-4621-0125

**Abstract:** Background: Pig slurry can negatively impact on the environmental, animal, and human health. Knowing the relationship between the organic and inorganic loads, pathogens, and toxicity allows identifying the main parameters to be removed or treated before final disposal. The aim of this study was to evaluate the relationships between the physicochemical properties, microbiological, and parasitological content, ecotoxicological effects, and biochemical methane potential (BMP) of pig slurries. Methods: Ten pig slurry samples at two production stages were characterized and a BMP test at two substrate/inoculum (S/I) ratios was conducted to compare the methane yields. Results: We found high content of Cu, Zn, quaternary ammonium, pathogenic microorganisms (*E. coli* and *Salmonella*), and parasites (*Trichuris* and *Trichostrongylus*). Toxicity on lettuce, radish, and *Daphnia* was observed with a slurry concentration greater than 1.35%. Positive correlations were found between toxicity on *Daphnia* and chemical oxygen demand (COD), sulfate, Zn, and Cu, as well as between phytotoxicity and COD, $NH_4$, Na, K, and conductivity. The lowest S/I ratio showed 13% more methane yield. It was associated with high removals of COD and volatile fatty acids. Conclusions: We recommend using a low S/I ratio to treat pig slurry as it improves the efficiency of the anaerobic process.

**Keywords:** pig manure; toxicity; pathogen; parasites; effluent; anaerobic digestion; waste management

## 1. Introduction

Forty percent of the meat consumed worldwide corresponds to pork. In Argentina, this production increased by 198.9% between 2002 and 2018 [1]. Pig production tends to intensify and the number of pig farms to increase, whereas in most of them, adequate slurry management strategies are not applied. Pig slurry is used worldwide as fertilizer due to the content of micro and macronutrients such as P, N, and organic matter [2,3]. Therefore, a key factor in the efficient use of slurry is to know its composition before applying a waste treatment or using it as crop fertilizer to avoid application rates over the crop requirements [4]. However, most of the farmers are generally not aware of the quality of their pig slurry, which also contains metals, hormones, antibiotics, radionuclides, and salts [5,6]. Nowadays, there is a raised concern for the increased survival of pathogenic and zoonotic agents in animal manure due to their possible transmission to other animals or humans [7,8]. Pig slurry is a liquid mixture of pig feces and urine that contains an abundance of pathogenic and non-pathogenic bacteria, viruses, and parasites [3,9]. The main intestinal parasites commonly observed in pigs are *Ascaris suum*, *Eimeria* spp., *Balantidium coli*, *Strongyloides*

*ransomi*, *Cryptosporidium* spp, *Oesophagostomum* spp., and *Trichuris suis* [10,11]. Enterobacteriaceae is the main bacteria family in pig slurry. Microorganisms and parasites can be transmitted through direct contact with manure or indirectly through the environment. Therefore, manure management must include a sanitization stage to reduce its negative impact on human and animal health [12]. Both nutrients and trace elements from pig manure may impact the environment [13]. N and P affect biodiversity due to surface and underground water eutrophication [14,15]. For example, the "Nitrate Directive" of the European Union (91/676/EEC) seeks to reduce water pollution caused by nitrates from agricultural sources. High concentrations of Cu and Zn in soil cause toxicity to animals, plants, and microorganisms [5,16]. Additionally, the negative impact on plant development by exposure to xenobiotic compounds introduced by wastewater and sludge land applications was studied [17]. Both seed germination and root elongation and *Daphnia magna* immobilization toxicity tests were successfully used for evaluating whole effluent toxicity [18,19]. Therefore, the combined use of toxicity tests and physicochemical parameters allows the integral assessment of the raw manure before disposal [19–22].

On the other hand, it is known that the variability of pig slurry characteristics depends on animal management, i.e., diet, animal age, housing, cleaning products, and slurry storage period [23–26]. Several manure treatment technologies are often used worldwide for pig slurry, such as anaerobic digestion [3,27], composting [5], and wetlands [28]. Furthermore, it creates new opportunities to better manage the nutrients and organic matter in agriculture [29,30]. For this, the slurry composition needs to be studied to apply the more appropriate treatment. In particular, anaerobic digestion represents a widely used process to reduce greenhouse gas emissions and water pollution and contributes to renewable energy supply by obtaining methane [27,31]. For planning and projection of an anaerobic treatment system, a priori determination of the methane potential of the organic substrate is imperative [32]. The biochemical methane potential (BMP) test is a useful laboratory-scale tool to assess methane yield. Therefore, we evaluated the relationships between the physicochemical properties, microbiological and parasitological content, ecotoxicological effects, and biochemical methane potential of pig slurries.

## 2. Materials and Methods

### 2.1. Pig Farm Management and Slurry Sampling

Pig slurry was obtained from an intensified farm located in Marcos Juarez, Argentina. This full-cycle farm had five production stages laid in different sheds: breeding pigs and maternity, pregnant sows, weaners (W), growing, and finishers (F). Diet was based on dry-feeding with soy pellets, flour corn, and vitamin and mineral supplement. The W and F stages were selected because they had sheds with fully slatted floor systems and different storage periods of pig slurry. Cleaning was manually performed with a hose and slurries were stored in a manure pit below the slatted floor. Slurry partial extractions were weekly made to control the volume of the liquid inside the manure pit. In W sheds, a complete extraction of slurry was made after one productive cycle (40 d), whereas in F sheds it was made after two productive cycles (100–120 d). Pig slurry samples were taken for 10 consecutive months at W and F sheds.

### 2.2. Chemicals

Potassium dichromate ($K_2Cr_2O_7$, Biopack, 99% purity, Argentina), zinc chloride ($ZnCl_2$, Biopack, 97% purity, Argentina), L-ascorbic acid ($C_6H_8O_6$, Biopack, 99% purity, Argentina), mercury sulfate ($HgSO_4$, Biopack, 98% purity, Argentina), silver sulfate ($Ag_2SO_4$, Biopack, 98% purity, Argentina), barium chloride ($BaCl_2$, Biopack, 99% purity, Argentina), hexadecyltrimethylammonium bromide (CTAB, Cat. No. 24010-66, Hach, Ames, IA, USA), and Hach quaternary ammonium compounds (QAC, Cat. No. 24012-68, Hach, USA).

### 2.3. Physicochemical, Microbiological, and Parasitological Analysis

Electrical conductivity (EC), pH, total (TS) and volatile solids (VS), total ($COD_T$) and soluble chemical oxygen demand ($COD_S$), biochemical oxygen demand at 5 d ($BOD_5$), total (TP) and soluble phosphorus (SP), total alkalinity (TA), sulfate ($SO_4^{2-}$), total Kjeldahl nitrogen (TKN), total ammonia nitrogen (TAN), cations ($Ca^{2+}$, $Mg^{2+}$, $K^+$, $Na^+$, $Cu^+$, $Fe^{2+}$, $Zn^{2+}$, $Mn^{2+}$), fecal coliforms, *Escherichia coli*, and *Salmonella* spp. were determined, according to APHA [33]. Quaternary ammonium compounds (quats) were measured by the direct binary complex method (HACH Method 8337) using a standard curve prepared with CTAB and Hach QAC reagents. The determination and quantification of helminth eggs were performed using the method proposed by Roberts and O'Sullivan [34]. All the parameters were determined by triplicate.

### 2.4. Toxicity Tests

One aquatic and two terrestrial species were used to assess the adverse effects of pig slurry. A total of 9 and 10 samples of pig slurry were assessed at W and F sheds, respectively.

#### 2.4.1. Daphnia Magna Immobilization Test

Acute toxicity tests were carried out using neonates of *Daphnia magna*, according to the US Environmental Protection Agency (USEPA) [35]. Experimental design consisted of 8 treatments (7 pig slurry concentrations and a negative control group) by triplicate for each production stage and sampling time. The pig slurry concentrations used in the tests were 0.1, 1, 3, 5, 9, 15, and 25% *v/v*. Ten neonates (<24 h of hatching) were exposed for 48 h in a static-flow system, containing 30 mL of sample or dilution water. Dechlorinated and aerated water (pH = $8.0 \pm 0.4$; EC = $627 \pm 49$ μS/cm; *n* = 5) was used as dilution water and for negative controls. Chromium ($K_2Cr_2O_7$) was used as reference toxic in positive controls in the following concentrations: 0.1, 0.25, 0.4, 0.55, and 0.7 mg/L. Experiments were conducted under controlled conditions ($23 \pm 2$ °C and 16:8-h light:dark). Toxicity endpoints assessed were effective concentration 50 ($EC_{50}$), LOEC (lowest observed effect concentration), and NOEC (no observed effect concentration). The quality controls used were carried out according to Young et al. [21]. For this, $EC_{50}$ values between 0.15 and 0.45 mg/L were used as reference in positive control, based on $\pm$ 2 SD obtained from internal control chart of the *D. magna* toxicity test (*n* = 20 tests).

#### 2.4.2. Seed Germination and Root Elongation Toxicity Test

Seeds of lettuce (*Lactuca sativa* L. variety "Criolla") and radish (*Raphanus sativus* variety "Puntas Blancas") without previous chemical treatment were provided by INTA (Argentina) and used for acute toxicity tests, according to Young et al. [36]. Experimental design consisted of 10 treatments (9 pig slurry concentrations and a negative control group) by triplicate for each production stage, sampling time, and plant species. The pig slurry concentrations used in the tests were 0.1, 0.4, 0.8, 1.6, 2, 5, 9, 14, and 18% *v/v*. Deionized water was used as dilution water and in negative controls. In addition, increasing concentrations of zinc ($ZnCl_2$) were used as positive controls: 18.75, 37.5, 75, 150, and 300 mg/L. The quality controls were carried out according to Young et al. [19]. For this, $IC_{50}$ values of root elongation between 21.6 and 89.2 mg/L for *L. sativa* and between 52.4 and 112.8 mg/L for *R. sativus* were used as reference in positive control, based on $\pm 2$ SD obtained from internal control charts of each toxicity test (*n* = 18 tests each).

Ten seeds were exposed to 4 mL of each treatment in 90-mm diameter Petri dishes lined with germination paper (Munktell AB Box 300, Grycksbo, Sweden). A total of 11,550 seeds of each species were used in these experiments. Toxicity endpoints assessed on seed germination and root elongation were inhibitory concentration 50 ($IC_{50}$), NOEC, and LOEC. Root length was used to calculate the relative growth index (RGI; Equation (1)), according to Young et al. [36]. RGI values between 0 and 0.8 indicate inhibition of root elongation, values higher than 0.8 and lower than 1.2 indicate no significant effect, and values higher than 1.2 indicate stimulation of root elongation [36]. The number of germinated seeds

and root length were used to calculate germination index (GI; Equation (2)), according to Zucconi et al. [37]. GI values lower than 80% indicate seed inhibition. The phytotoxicity indexes $RGIC_{0.8}$ and $GIC_{80\%}$ were used to compare the toxicity of pig slurries from F and W sheds, according to Young et al. [21]. These authors defined that $RGIC_{0.8}$ and $GIC_{80\%}$ estimate the lowest concentration to get inhibition of root elongation (RGI = 0.8) and a response of 80% in GI, respectively. Values of the $RGIC_{0.8}$ and $GIC_{80\%}$ were differentiated into two categories [19]: (a) inhibitory effects: $\leq 100\%$; and (b) non-inhibitory effects: >100%.

$$RGI = \frac{RLE}{RLC} \qquad (1)$$

$$GI\ (\%) = \frac{GSE}{GSC} \times \frac{RLE}{RLC} \times 100 \qquad (2)$$

where RLE is the average root length in the slurry (mm), RLC is the average root length in the control (mm), GSE is the average number of germinated seeds in the slurry, and GSC is the average number of germinated seeds in the control.

### 2.5. Biochemical Methane Potential (BMP)

BMP test was conducted mixing an aliquot of the inoculum, pig slurry (substrate), and mineral solution, following the recommendations and criteria described by Holliger et al. [38]. Pig slurry of F sheds was selected for this test due it showed greater organic matter content than W. The inoculum was collected from an agroindustrial waste biogas plant. A specific methanogenic activity test (SMA) was conducted to ensure its activity. SMA was 0.08 g COD/g VSS d, and the inoculum characteristics were: 35.7 g TSS/ L, 24.4 g VSS/ L, and 44.7 g COD/L.

A completely randomized experimental design consisted of 3 treatments by triplicate. Two S/I ratios of 0.36 (T1) and 0.62 g COD/g VSS (T2) were evaluated. These ratios were achieved with equal inoculum mass but different substrate mass. The inoculum was diluted at 7 g SSV/L with mineral solution, according to the procedure proposed by Angelidaki et al. [39]. A control group (inoculum + mineral solution) was used to determine the endogenous methane production. Each batch reactor was considered as the experimental unit, which consisted of glass bottles (Schott-Duran) with a total volume of 560 mL and an effective volume of 448 mL, inoculum, substrate, and mineral solution (Table S1). The mixture in the reactor was adjusted to pH 7.0. Each reactor had 20% of headspace and a total volume of 487 mL that was flushed with nitrogen to remove oxygen. Then, reactors were placed in an incubator at $35 \pm 1$ °C and mixed by swirling manually for 30 s twice a day. The BMP test was finished on day 50, based on the 1% criterion which establishes a production rate <1% of net production per day for at least 3 d [38].

Biogas production was measured by a manometric method. Methane and carbon dioxide contents in biogas were quantified by gas chromatography (Hewlett Packard 5890 GC System), according to Bres et al. [19]. $COD_T$, $COD_S$, NTK, TS, VS, TAN, free ammonia nitrogen (FAN), volatile fatty acids (VFA), partial alkalinity (PA), and TA were determined at the initial and final sampling time of the BMP test. VFA, PA, and TA were determined according to Jenkins et al. [40], and FAN concentration was calculated following the procedure of Hansen et al. [41].

### 2.6. Data Analysis

Physicochemical properties of pig slurry from W and F stages were analyzed by Kruskal–Wallis test. NOEC and LOEC were determined by one-way ANOVA and Dunnet's post hoc test ($p < 0.05$). In BMP test, parameters were analyzed by two-way analysis of variance (ANOVA) followed by Bonferroni contrast. Complementary, principal component analyses (PCA), and Spearman correlation analyses were conducted between physicochemical parameters and ecotoxicological endpoints or biogas yield parameters. Data analyses were performed using GraphPad Prism and InfoStat software.

## 3. Results

### 3.1. Pig Slurry Monitoring

3.1.1. Physicochemical Properties

Physicochemical parameters measured in pig slurries from both sheds are shown in Table 1. Although results showed high EC values, no significant differences ($p > 0.05$) between samples from W and F stages were observed. A high concentration of TA, a parameter associated with EC, was observed in samples from F and W sheds.

**Table 1.** Mean ($\pm$SD) physicochemical and microbiological parameters of pig slurry ($n = 10$).

| Parameter | Unit | Weaners (W) | Finishers (F) |
|---|---|---|---|
| pH | | 6.5 $\pm$ 0.3 [a] | 6.3 $\pm$ 0.3 [a] |
| EC | mS/cm | 14.7 $\pm$ 4.1 [a] | 13.5 $\pm$ 3.4 [a] |
| TA | g CaCO$_3$/L | 6.1 $\pm$ 1.3 [a] | 6.1 $\pm$ 1.9 [a] |
| TS | g/L | 13.0 $\pm$ 3.2 [a] | 18.0 $\pm$ 10.7 [a] |
| VS | g/L | 7.6 $\pm$ 2.2 [a] | 12.9 $\pm$ 9.8 [a] |
| VS/TS | | 58.4 $\pm$ 8.5 [a] | 66.2 $\pm$ 13.5 [a] |
| TP | mg/L | 238.1 $\pm$ 93.4 [a] | 423.1 $\pm$ 110.2 [a] |
| SP | mg/L | 85.6 $\pm$ 43.8 [a] | 122.0 $\pm$ 57.8 [a] |
| COD$_T$ | g/L | 27.7 $\pm$ 18.0 [a] | 33.1 $\pm$ 13.7 [a] |
| COD$_S$ | g/L | 18.9 $\pm$ 8.0 [a] | 20.5 $\pm$ 8.2 [a] |
| BOD$_5$ | g/L | 13.5 $\pm$ 5.9 [a] | 20.3 $\pm$ 9.6 [a] |
| TKN | % | 0.22 $\pm$ 0.08 [a] | 0.24 $\pm$ 0.08 [a] |
| NH$_4^+$ | % | 0.20 $\pm$ 0.06 [a] | 0.18 $\pm$ 0.06 [a] |
| SO$_4^{2-}$ | mg/L | 385.0 $\pm$ 185.6 [a] | 365.0 $\pm$ 133.4 [a] |
| Ca$^{2+}$ | mg/L | 263.2 $\pm$ 64.8 [a] | 266.6 $\pm$ 65.6 [a] |
| Cu$^+$ | mg/L | 1.9 $\pm$ 0.4 [a] | 1.9 $\pm$ 0.9 [a] |
| K$^+$ | mg/L | 1648.6 $\pm$ 466.0 [a] | 1451.4 $\pm$ 323.7 [a] |
| Zn$^{2+}$ | mg/L | 1.3 $\pm$ 0.2 [a] | 2.1 $\pm$ 1.3 [a] |
| Na$^+$ | mg/L | 932.8 $\pm$ 260.5 [a] | 809.2 $\pm$ 151.1 [a] |
| Mg$^{2+}$ | mg/L | 122.0 $\pm$ 55.5 [a] | 146.2 $\pm$ 57.6 [a] |
| Mn | mg/L | 0.6 $\pm$ 0.3 [a] | 2.6 $\pm$ 1.9 [b] |
| Quats | mg CTAB/L | 75.0 $\pm$ 6.6 [a] | 49.2 $\pm$ 15.1 [b] |
| Fecal coliforms | MPN/mL | $8.40 \times 10^{3}$ [a] | $9.96 \times 10^{3}$ [a] |
| *Escherichia coli* | MPN/mL | $8.40 \times 10^{3}$ [a] | $9.50 \times 10^{3}$ [a] |
| *Salmonella* spp. | presence | negative | positive |

MPN: most probable number. W: weaners shed, F: Finishers shed. SD: standard deviation. Different letters indicate significant differences ($p \leq 0.05$) between sheds.

In addition, a high organic and inorganic load was found. Although the highest concentrations of TS, VS, TP, COD$_T$, and COD$_S$ were recorded in F, no significant differences ($p > 0.05$) were observed between production stages. However, the highest VS/TS percentage in F shows that it had more organic matter than W (71.7% for F and 58.5% for W). It could lead to more potential of biogas and methane in F. Besides, these parameters showed a high variability during all the monitoring of pig slurries. Organic matter showed high concentrations for both stages (27.8 $\pm$ 6.5 and 33.1 $\pm$ 9.0 g COD$_T$/L for W and F, respectively). Moreover, the difference between COD$_T$ and COD$_S$ indicated that 68 and 62% of organic matter were solubilized in W and F slurries, respectively. The phosphorus concentration was in a range of 140 to 550 mg/L, showing a tendency to higher concentrations in F sheds. However, no significant differences ($p > 0.05$) between samples from W and F stages were observed. Furthermore, the cations showed high concentrations such as Na (932.8 $\pm$ 260.5 mg/L for W and 809.2 $\pm$ 151.1 mg/L for F) and K (1648.6 $\pm$ 466.0 mg/L for W and 1451.4 $\pm$ 323.7 mg/L for F). Particularly, Mn concentrations were higher in F sheds than in W sheds ($p \leq 0.05$).

Regarding nitrogenous compounds, we observed that the major proportion of nitrogen was in an inorganic form, mainly as TAN. In samples from W and F sheds, TAN represented 91 and 75% of TKN, respectively. In addition, sulfate presented a similar average

concentration for W and F sheds, showing the COD/sulfate ratio obtained was higher than 10. The highest concentration of quats was observed in the W stage ($p \leq 0.05$). It could be explained because W sheds (each 40 d) were disinfected with quats more frequently than F sheds (each 100–120 d).

### 3.1.2. Microbiological and Parasitological Characterization

The presence of fecal coliforms, *E. coli*, and *Salmonella* spp. of pig slurry from both sheds is shown in Table 1. Fecal coliforms and *E. coli* count (MPN/mL) were similar in both production stages. However, *Salmonella* spp. was not detected in any of the samples from the W stage, whereas 60% of the samples from the F stage had a presence of this pathogen.

Regarding parasitological analysis, helminth eggs were found in both production stages, mainly represented by *Trichuris* spp. and *Trichostrongylus* spp. The occurrence of positive samples was 60% in the F stage and 100% in the W stage.

### 3.1.3. Ecotoxicity

The quality controls at each test organism were acceptable according to the criteria established; even values obtained in the positive controls were within the acceptable $EC_{50}$ or $IC_{50}$ range of the internal control charts. In negative controls, coefficients of variation of root elongation were $14.1 \pm 2.1\%$ and $17.4 \pm 3.1\%$, and the percentages of germinated seeds were $97.3 \pm 3.9\%$ and $98.1 \pm 3.0\%$ for *L. sativa* and *R. sativus*, respectively. Additionally, the survival of *D. magna* neonates was $97.9 \pm 5.9\%$ in negative controls.

All samples of pig slurry caused toxicity to *D. magna*, *L. sativa,* and *R. sativus*. Ecotoxicological endpoints did not show significant differences ($p > 0.05$) between pig slurries from W and F sheds (Table 2). Standard deviation values of the several endpoints indicate a low variability of toxicity during all the monitoring of pig slurries. The sensitivity measured in terms of $EC_{50}$ or $IC_{50}$ was highest for *D. magna*, followed by lettuce, and then radish.

**Table 2.** Mean (±SD) ecotoxicological endpoints of the test organisms at each sampled pig slurry. The lower and upper limits of the 95% confidence interval are shown in parentheses.

| Species | Endpoint | Weaners (W) | Finishers (F) |
|---|---|---|---|
| *Daphnia magna* | *Immobilization* | | |
| | $EC_{50}$ (%) | $2.73 \pm 0.84$ (2.45–3.01) | $2.16 \pm 0.89$ (1.89–2.43) |
| | LOEC (%) | $3.73 \pm 1.93$ (3.09–4.38) | $2.35 \pm 1.77$ (1.81–2.88) |
| | NOEC (%) | $1.60 \pm 0.69$ (1.37–1.83) | $1.24 \pm 0.68$ (1.03–1.44) |
| *Lactuca sativa* | *Root elongation* | | |
| | $IC_{50}$ (%) | $6.00 \pm 9.44$ (2.43–9.56) | $5.40 \pm 3.99$ (3.98–6.81) |
| | LOEC (%) | $4.23 \pm 6.86$ (1.63–6.82) | $3.05 \pm 3.06$ (1.97–4.13) |
| | NOEC (%) | $2.86 \pm 5.23$ (0.88–4.84) | $1.53 \pm 1.61$ (0.96–2.10) |
| | *Phytotoxicity indexes* | | |
| | $RGIC_{0.8}$ (%) | $2.55 \pm 4.09$ (1.01–4.10) | $2.23 \pm 2.42$ (1.37–3.08) |
| | $GIC_{80\%}$ (%) | $2.51 \pm 4.06$ (0.98–4.05) | $2.07 \pm 2.36$ (1.24–2.91) |
| *Raphanus sativus* | *Root elongation* | | |
| | $IC_{50}$ (%) | $7.17 \pm 10.40$ (3.24–11.10) | $7.03 \pm 6.80$ (4.46–9.59) |
| | LOEC (%) | $3.80 \pm 5.47$ (1.73–5.87) | $1.74 \pm 2.11$ (0.99–2.48) |
| | NOEC (%) | $2.17 \pm 3.50$ (0.85–3.50) | $0.72 \pm 0.86$ (0.41–1.02) |
| | *Phytotoxicity indexes* | | |
| | $RGIC_{0.8}$ (%) | $2.28 \pm 3.63$ (0.91–3.66) | $1.40 \pm 1.83$ (0.71–2.09) |
| | $GIC_{80\%}$ (%) | $2.27 \pm 3.70$ (0.87–3.67) | $1.35 \pm 1.86$ (0.64–2.05) |

Statistical analysis did not show significant differences ($p > 0.05$) between slurries for any tested endpoint. A total of 100% of the samples exhibited a toxic response. SD: standard deviation.

### 3.1.4. Correlation and Principal Component Analyses

Multivariate analyses indicated a relationship between toxicity and the inorganic and organic content. PCA accounts for 73.1% of the variability of the data matrix (Figure 1). This analysis showed a positive association of TAN with pH and Na.

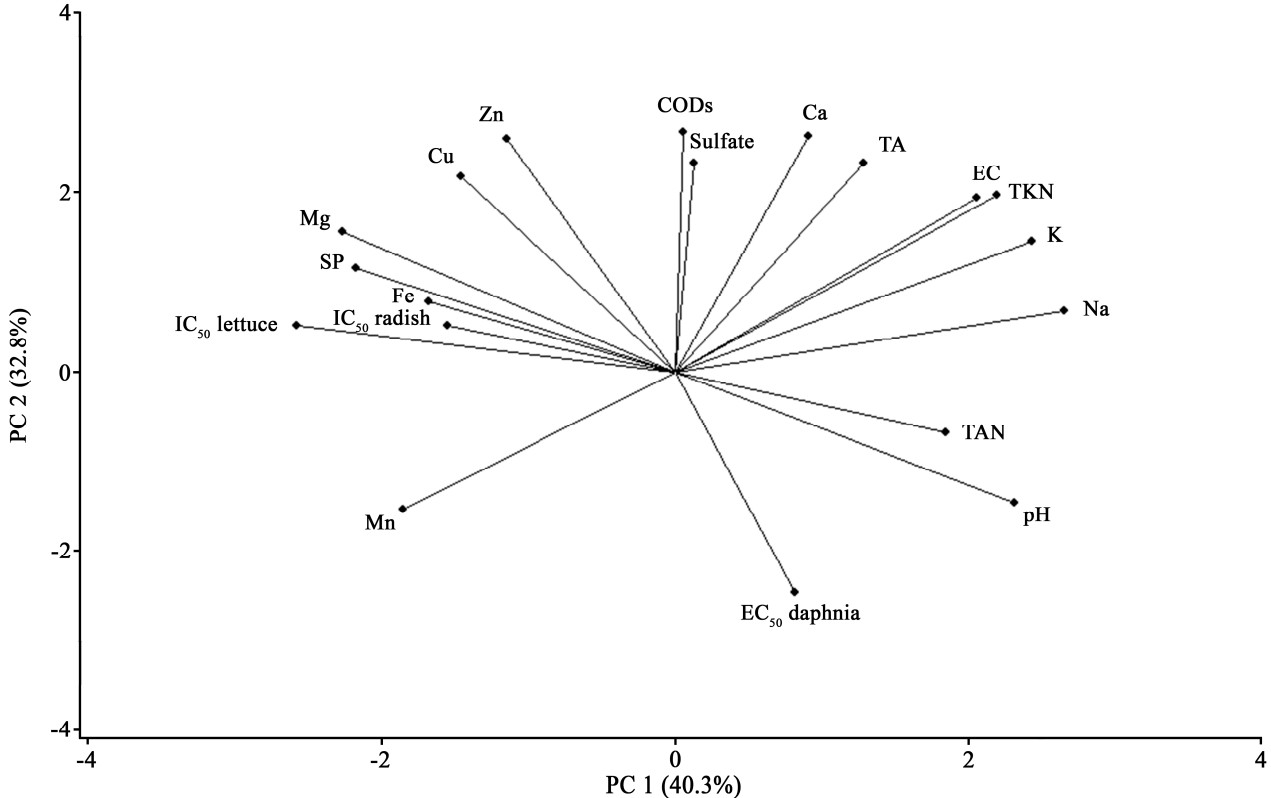

**Figure 1.** Principal components analysis (PCA) shows the association between physicochemical parameters and ecotoxicological endpoints.

Moreover, EC was associated positively with TKN, K, TA, and Na, whereas negatively with Mn. In addition, the Spearman correlation showed a positive association between TKN and EC (R = 0.89; $p \leq 0.01$; Table 3).

Regarding the multivariate analyses between physicochemical and ecotoxicological parameters, PCA showed a negative association between $EC_{50}$ of *D. magna* and $COD_S$, sulfate, Zn, and Cu. Moreover, Spearman coefficients showed that $EC_{50}$ of *D. magna* correlated negatively to $COD_S$ (R = −0.70; $p \leq 0.01$) and sulfate (R = −0.70; $p \leq 0.01$). PCA showed a positive association of $IC_{50}$ of *R. sativus* and *L. sativa* with SP, Fe, and Mg, but showed a negative association with TAN, Na, K, TKN, pH, and EC. In addition, $IC_{50}$ of *L. sativa* correlated negatively with Na (R = −0.73; $p \leq 0.05$), and positively with Mg (R = 0.76; $p \leq 0.05$).

We found that the phytotoxicity indexes ($RGIC_{0.8}$ and $GIC_{80\%}$) estimated for lettuce correlated positively with SP (R = 0.73 and 0.74, respectively; $p \leq 0.05$) and Mg (R = 0.84 for both indexes; $p \leq 0.01$). Spearman coefficients showed a positive correlation between $IC_{50}$ of *R. sativus* and $COD_T$ (R = 0.77; $p \leq 0.01$), whereas showed a negative correlation between quats and *R. sativus* and *L. sativa* (R = 0.72). Additionally, we observed a positive correlation between the $IC_{50}$ of *R. sativus* and the $IC_{50}$ of *L. sativa* (R = 0.84; $p \leq 0.01$). This correlation indicates that both plant species had a similar toxic response.

**Table 3.** Correlation coefficients among ecotoxicological and physicochemical parameters measured at pig slurry samples ($n = 14$).

| | | D. magna | L. sativa (Lettuce) | | | R. sativus (Radish) | | | | Physicochemical Parameters | | | | | | | |
|---|---|---|---|---|---|---|---|---|---|---|---|---|---|---|---|---|---|
| | | LOEC | LOEC | RGIC$_{0.8}$ | GIC$_{80\%}$ | IC$_{50}$ | LOEC | RGIC$_{0.8}$ | GIC$_{80\%}$ | SP | COD$_S$ | SO$_4^-$ | TA | TKN | Mg | Na | Cu |
| *D. magna* | EC$_{50}$ | 0.85 ** | ns | ns | ns | ns | ns | ns | ns | ns | −0.70 ** | −0.70 ** | ns | ns | ns | ns | ns |
| | NOEC | 0.91 ** | ns | ns | ns | ns | ns | ns | ns | ns | ns | ns | ns | ns | ns | ns | ns |
| | LOEC | | ns | ns | ns | ns | ns | ns | ns | ns | ns | ns | −0.70 ** | ns | ns | ns | ns |
| *L. sativa* | IC$_{50}$ | | 0.92 ** | 0.95 ** | 0.95 ** | 0.84 ** | ns | ns | ns | ns | ns | ns | ns | ns | 0.76 * | −0.73* | ns |
| | NOEC | | 0.98 ** | 0.94 ** | 0.94 ** | 0.78 ** | ns | ns | ns | ns | ns | ns | ns | ns | 0.75 * | ns | ns |
| | LOEC | | | 0.95 ** | 0.95 ** | 0.78 ** | ns | ns | ns | ns | ns | ns | ns | ns | 0.74 * | ns | ns |
| | RGIC$_{0.8}$ | | | | 1.00 ** | 0.79 ** | ns | ns | ns | 0.73 * | ns | ns | ns | ns | 0.84 ** | ns | ns |
| | GIC$_{80\%}$ | | | | | 0.79 ** | ns | ns | ns | 0.74 * | ns | ns | ns | ns | 0.84 ** | ns | ns |
| *R. sativus* | IC$_{50}$ | | | | | | 0.75 ** | 0.84 ** | 0.81 ** | ns | ns | ns | ns | ns | ns | ns | ns |
| | NOEC | | | | | | 0.99 ** | 0.95 ** | 0.96 ** | ns | ns | ns | ns | ns | ns | ns | ns |
| | LOEC | | | | | | | 0.98 ** | 0.98 ** | ns | ns | ns | ns | ns | ns | ns | ns |
| | RGIC$_{0.8}$ | | | | | | | | 1.00 ** | ns | ns | ns | ns | ns | ns | ns | ns |
| Physicochemical parameters | pH | | | | | | | | | −0.72 ** | ns | | | | | | |
| | EC | | | | | | | | | ns | 0.80** | ns | 0.92** | 0.89** | | | |
| | TS | | | | | | | | | | ns | ns | ns | ns | ns | | |
| | COD$_S$ | | | | | | | | | ns | | | 0.84** | 0.80** | | | |
| | TA | | | | | | | | | | | | | 0.91** | | | |
| | Ca | | | | | | | | | | | | | | 0.87 ** | 0.98 ** | |
| | Mg | | | | | | | | | | | | | | | 0.77 ** | |
| | K | | | | | | | | | | | | | | | 1.00** | |
| | Zn | | | | | | | | | | | | | | | | 0.88 ** |

* indicates significant correlations at $p \leq 0.05$; ** indicates significant correlations at $p \leq 0.01$; ns: not significant.

### 3.2. Biodegradability: Biogas and Methane Production

In the BMP test, the cumulative production of methane was $408 \pm 9$ and $580 \pm 44$ mL for T2 and T1, respectively (Figure S1). No significant differences ($p > 0.05$) were observed between treatments for this parameter. The methane yields obtained were $0.25 \pm 0.05$ L $CH_4$/g $VS_{added}$ and $0.21 \pm 0.02$ L $CH_4$/g $VS_{added}$ for T1 and T2, respectively. T1 showed 13% more methane yield than T2, although no significant differences were observed between treatments ($p \leq 0.05$) (Figure 2). Better performance was found with low pig slurry load, i.e., T1 was greater than T2 from day 13 to the end of the test. Although the difference in $COD_T$ values between treatments was double, the difference in $COD_S$ represented a higher proportion (Table 4).

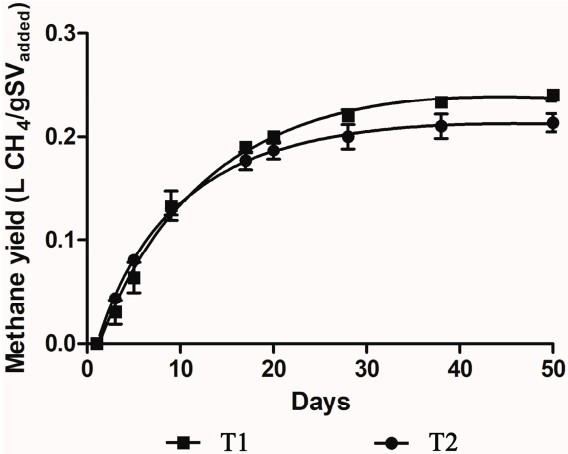

**Figure 2.** Mean ($\pm$SD) methane yield for T1 and T2 ($n = 3$). T1: substrate/inoculum ratio of 0.36 g COD/g VSS, T2: substrate/inoculum ratio of 0.62 g COD/g VSS.

**Table 4.** Mean ($\pm$SD) physicochemical parameters of pig slurry at initial and final sampling time of the BMP test ($n = 3$).

| Parameter | Pig Slurry | Initial | | Final | |
|---|---|---|---|---|---|
| | | T1 | T2 | T1 | T2 |
| pH | $6.8 \pm 0.1$ | $6.9 \pm 0.0$ [a] | $7.0 \pm 0.0$ [a] | $7.1 \pm 0.1$ * | $7.2 \pm 0.0$ * |
| $COD_T$ (g/L) | $32.9 \pm 1.8$ | $13.2 \pm 0.6$ [a] | $24.2 \pm 0.6$ [b] | $7.4 \pm 0.3$ * | $15.2 \pm 0.7$ * |
| $COD_S$ (g/L) | nd | $4.6 \pm 0.6$ [a] | $12.6 \pm 0.6$ [b] | $0.8 \pm 0.2$ * | $3.3 \pm 0.7$ * |
| VFA (g/L) | $2.6 \pm 0.1$ | $0.5 \pm 0.0$ [a] | $0.7 \pm 0.0$ [b] | $0.05 \pm 0.0$ * | $0.08 \pm 0.0$ * |
| PA (g/L) | $2.6 \pm 0.0$ | $0.8 \pm 0.0$ [a] | $1.3 \pm 0.0$ [b] | $1.3 \pm 0.0$ * | $3.5 \pm 0.1$ * |
| TA (g/L) | $6.2 \pm 0.0$ | $1.3 \pm 0.0$ [a] | $2.0 \pm 0.0$ [b] | $1.7 \pm 0.1$ * | $5.2 \pm 0.0$ * |
| TAN (g/L) | $3.8 \pm 0.0$ | $1.8 \pm 0.0$ [a] | $2.7 \pm 0.0$ [b] | $2.7 \pm 0.0$ * | $3.2 \pm 0.0$ * |
| FAN (mg/L) | nd | $18.6 \pm 0.4$ [a] | $35.0 \pm 0.5$ [b] | $39.2 \pm 0.6$ * | $63.7 \pm 0.7$ * |
| $COD_T$ removal (%) | nd | nd | nd | $40.8 \pm 3.7$ [a] | $37.2 \pm 3.9$ [a] |
| $COD_S$ removal (%) | nd | nd | nd | $81.5 \pm 4.2$ [a] | $66.6 \pm 6.9$ [b] |
| VFA removal (%) | nd | nd | nd | $89.3 \pm 0.8$ [a] | $88.3 \pm 0.9$ [a] |
| TAN increasement (%) | nd | nd | nd | $48.4 \pm 3.1$ [a] | $19.1 \pm 0.6$ [b] |
| FAN increasement (%) | nd | nd | nd | $110.0 \pm 4.2$ [a] | $82.2 \pm 0.3$ [b] |

T1: S/I ratio of 0.36 g COD/g VSS, T2: S/I ratio of 0.62 g COD/g VSS. nd: no determined. SD: standard deviation. Different letters indicate significant differences between treatments at initial or final sampling time of the test ($p \leq 0.05$). Asterisks indicate significant differences between the initial and final sampling times of the test for each treatment ($p \leq 0.05$).

On the other hand, PCA showed that T1 was associated with high methane yield and removals of $COD_T$, $COD_S$, and VFA, whereas T2 was associated with high values of methane cumulated, $COD_T$, $COD_S$, TA, VFA, TAN, and FAN (Figure S2). Additionally, methane yield correlated negatively with VFA ($R = -1$; $p \leq 0.05$), and pH correlated positively with TAN and FAN (both $R = 0.94$; $p \leq 0.05$).

## 4. Discussion

In this study, the physicochemical characteristics of pig slurries analyzed were similar to previous studies [2,24,42]. The high variability observed in several parameters during all the monitoring could be related to the management practices and long storage periods in pits because water volume used for cleaning sheds modifies density, mineral concentration, and dry matter [43,44]. The cleaning of sheds was performed manually, causing a discontinuous water flow to the system.

The high nutrient and organic matter concentrations observed during all the monitoring of pig slurries indicated favorable conditions for a biological process [19]. Besides, values of $COD_T$, $BOD_5$, TKN, TAN, and cations (Na and K) were similar to those reported by other authors, who also studied pig slurry composition in confinement animal feeding operations (CAFOs) [2,25,44]. The high solubility of organic matter found in our study is also attributable to slurry management due to the long storage periods before disposal. In addition, high EC values could be explained by the high content of salts, protein, and ammonium in food [25]. Other authors have reported similar EC values in pig slurry [2,24,42]. The use of pig slurry as a soil fertilizer without a percolating water regime could cause salinization due to high EC [42]. In the present study, a positive correlation between EC and TKN was found ($R = 0.89$, $p < 0.01$), which was similar to other studies that evaluated effluents [19,24]. Moreover, the high EC values are in concordance with the high concentrations of TA. For example, Villamar et al. [45] found similar values of alkalinity on pig slurries. Moral et al. [25] also reported pH values ranging from 6 to 7 in pig farms in Southeast Spain. The pH of urine is affected by the dietary electrolyte balance [46]. Moreover, the pH of the slurry can be influenced by high concentrations of VFA, as were found in the BMP test of the present study. These compounds are released during the organic matter degradation for a long storage time. The anaerobic process is sensitive to pH fluctuations in the system, and especially methanogens require a strict pH range (6.5–7.2). Therefore, slurries with a high concentration of alkalinity generate a buffering capacity of carbonates and bicarbonates, minimizing variations in the pH when VFA are produced in anaerobic reactors [40].

Pigs excrete a high percentage of the nitrogen consumed in their diet. In this sense, we observed that the largest proportion of nitrogen was in an inorganic form, mainly as TAN. It is known that slurries stored for long periods in anaerobic condition reach a high mineralization degree [2,24,43]. Ammonium ($NH_4^+$) can also be toxic to microbes in biological treatment systems [47]. Particularly, the non-ionized form of Nitrogen ($NH_3$) is more toxic than the ionized form ($NH_4^+$) in anaerobic conditions, since it can penetrate the cell wall [48]. Therefore, special attention should be considered when anaerobic digestion is proposed as a treatment system for pig slurry. Nonetheless, there is evidence that $NH_4^+$ is toxic to aquatic organisms like *D. magna* when pig slurry is discharged into freshwater [45].

Regarding cations, results are consistent with those reported by other authors [49], although Moral et al. [25] found higher concentrations of K in the F stage. Several cations such as K, Na, Cu, Zn, and Mn are used as a supplement in diets to improve growth rate and prevent possible symptoms due to the deficiency of these ions, even in concentrations that exceed physiological requirements [14,50]. For instance, it is estimated that pigs excrete approximately 66% of Na and 59% of K consumed in their diet [51]. Similarly, Clemente et al. [5] observed high concentrations of Cu and Zn when they evaluated the separated solid fraction of pig slurries. Cu and Zn are important trace elements fulfilling many metabolic functions in animals and, at the same time, are associated with environmental pollution [7]. Mantovi et al. [52] evaluated the correlation between the metal content in soil and the extent of manure application. These authors found that Cu and Zn concentrations increased after soil application of liquid manure. In the present study, the highest Mn concentrations were found in F sheds. Although there are studies on the effects of Mn in the diet of growing-finishing pigs on growth rate [50], there is little information on the Mn concentration released to the slurry. Additionally, phosphorus content in pig slurry is mainly bound to suspended solids [25]. The concentrations of TP measured in this study were similar to those reported by Martínez-Suller et al. [24], who also observed higher

concentrations in slurries from F sheds. This could be associated with the higher content of phosphorus in the diet provided during this production stage. Pigs excrete phosphorus as organic complexes such as phytic acid [51]. Phosphorus in the form of phytate is not available to non-ruminant animals. Therefore, unabsorbed phosphorus passes through the gastrointestinal tract, increasing its concentration in manure [51].

Sulfate is an important parameter for anaerobic digestion because is converted to hydrogen sulfide that inhibits methanogens, increasing the effect when low values of the COD/sulfate ratio are recorded [53]. For this reason, it is critical to evaluate sulfate concentrations in slurry. Villamar et al. [45] reported sulfate concentrations in the same range for pig slurry as those found in the present study. Additionally, the values of the COD/sulfate ratio obtained in this study were higher than 10, indicating no inhibition by hydrogen sulfide according to Shayegan et al. [53].

Quats are a cationic substance widely used for disinfection in animal production [54]. The highest concentration of quats in the W stage is attributable to the use frequency of this disinfectant in the farm, where W sheds were more frequently cleaned with quats (each 40 d) than F sheds (each 100–120 d). Li et al. [55] reported high concentrations of quats in sludge samples that were highly adsorbed to organic matter. Therefore, the use of amendments with quats causes a potential risk to terrestrial and aquatic organisms [17,54]. Additionally, some studies found a relationship between the presence of quats and the loss of methanogenic activity in anaerobic processes [56]. For example, Fernandez-Bayo et al. [57] found negative effects from 100 mg/L of quats on methane production. Lower concentrations of quats were found in the present study in comparison with those inhibitory concentrations of the anaerobic process reported by Fernandez-Bayo et al. [57].

Regarding microbiological monitoring, we found lower values of fecal coliforms than those reported by Grudziński et al. [12]. *Salmonella* spp. was not detected in any of the samples from the W stage, whereas was detected in the F stage. This could be attributable to the lower disinfection frequency of the F sheds. To illustrate this, Mannion et al. [58] showed that frequent cleaning and disinfection practices in pig farms are effective in significantly reducing Enterobacteriaceae levels. Moreover, the parasitological analysis showed the presence of helminth eggs, mainly represented by *Trichuris* spp. and *Trichostrongylus* spp. These zoonotic parasites are frequently found in pig slurry [7,10,11,59]. Recently, other authors also reported the presence of *Ascaris suum*, *Eimeria* spp., *Balantidium coli*, *Strongyloides ransomi*, *Cryptosporidium* spp, and *Oesophagostomum* spp. in the intestine of pigs [10,11]. Additionally, the highest observed frequency of parasites found in W sheds could be attributed to younger animals that have an immature immune system and, consequently, an increased risk of infection by parasites. Because pig slurry contains several species of pathogenic microorganisms and parasites, it should be treated before final disposal. The prevalence and concentration of pathogens in slurry depend on animal health, manure moisture, slurry management, and ambient temperature [49,60]. In addition, Plachá et al. [61] reported that high temperature, dryness, and UV light decrease their survival in fertilized soils. Authors observed ecological risks caused by pathogens and antibiotics after soil application [8,60].

In this study, the ecotoxicity of pig slurry was evaluated using *D. magna*, *L. sativa*, and *R. sativus*, and all the samples showed toxicity to the exposed organisms. The sensitivity measured in terms of $EC_{50}$ or $IC_{50}$ is coincident with those reported by Young et al. [21], who evaluated the toxicity of aqueous extracts from poultry manure and compost. When evaluating multivariate analysis to determine the relationship between toxicity and the inorganic and organic content, a positive association between TKN and EC was found. Similarly, other studies have also shown that EC correlates positively with TAN and TKN in animal effluents [19,24]. EC in slurries is affected by the concentration of the major cations dissolved in liquid phase (Na, K, Ca, Mg, $NH_4^+$). As most monovalent salts are almost completely dissociated in water, and K and $NH_4^+$ are the dominant cations in pig slurry, they are often correlated to EC [26]. Furthermore, the PCA analyses showed a negative association between $EC_{50}$ of *D. magna* and $COD_S$, sulfate, Zn, and Cu. Other authors have reported similar correlations with those found in this study. For example,

Fjällborg et al. [62] also found toxicity in *D. magna* exposed to metals, such as Zn, Ag, Cu, Fe, and Mn. In addition, Pablos et al. [63] studied the correlation between the physicochemical parameters of landfill leachate and toxicological endpoints on *D. magna*. These authors reported that the samples with high levels of TAN, TA, COD, and chloride should be directly classified as potentially toxic. Tigini et al. [64] studied the effects of pig slurry digestate on 7 species, including *D. magna*. They demonstrated that the high sensitivity of *D. magna* was related to TAN concentration, COD, and EC. Regarding the effects on *R. sativus* and *L. sativa*, a negative association with TAN, Na, K, TKN, pH, and EC was found. Indeed, Pampuro et al. [65] reported a significant negative correlation between EC and relative seed germination (GSE/GSC), evaluating the phytotoxicity of pig slurry-derived compost on *Lepidium sativum*. According to these authors, salinity can have a detrimental effect on seed germination and plant growth, especially during seedling development. In this sense, Halder et al. [66] observed a negative correlation between EC and GI. Additionally, a positive correlation between $COD_T$ and $IC_{50}$ of *R. sativus* in the present study was found. This is consistent with results published by several authors, who also reported correlations between phytotoxicity endpoints and COD from pulp and paper effluents [18] and anaerobic digestate [36]. In particular, Tigini et al. [64] suggested that phytotoxicity observed in the green algae *Raphidocelis subcapitata*, the aquatic plant *Lemna minor*, and the terrestrial plants *Cucumis sativus* and *Lepidium sativum* exposed to pig slurry digestate could be associated with high values of ammonium concentration, EC, and COD. Besides, quats correlated negatively with *R. sativus* and *L. sativa*. For example, Di Nica et al. [67] observed the inhibition in the growth of roots and shoots of wheat seedlings when applying concentrations of quats $\geq 1$ mg/L. In the same study, they also detected that long-term exposure to quats resulted in the browning of leaf edges and eventually chlorosis.

On the other hand, the BMP test had similar behavior to those results reported by Kafle and Kim [68] in batch experiments at a 0.5 S/I ratio of pig slurry. In addition, better performance of the anaerobic process in the present study using a low pig slurry load was observed. Similar results were obtained by Chae et al. [69], who reported lower methane yield with greater volume of pig manure ($0.32 \pm 0.01$ and $0.23 \pm 0.02$ L $CH_4$/g $VS_{added}$ for 20 and 40% *v/v* of pig manure at 35 °C, respectively). A negative correlation between methane yield and VFA was found, whereas an association between cumulative methane production and VFA was observed. Therefore, high substrate amounts cause an increase in the availability of easily hydrolyzable material, which in turn leads to VFA accumulation [70]. For example, Hobbs et al. [71] found that a high S/I ratio resulted in greater volumetric methane production and VFA accumulation. This accumulation induced a decrease in the pH value, causing a long lag period and/or anaerobic digestion inhibition. Alternatively, Mansour et al. [72] reported that a higher S/I ratio induces a decrease in the hydrolysis constant, indicating a reduction in the effectiveness of the anaerobic process. The implementation of biogas technology is visualized as an important factor in increasing renewable energy.

Anaerobic digestion, an environmental-friendly and effective treatment for pig slurry, has been widely used in several countries including China [30,73]. Particularly, in 2015, the Argentine government enacted a law that established the target of achieving 20% renewable energy by 2025 (Law no. 27,191). Consequently, the interest in biogas technology from pig slurry has increased among farmers and researchers. The biogas technology can reduce odor emission and COD by disposing of pig slurry in a closed system and producing biogas and fertilizer. Biogas can be used as renewable energy to generate heat and electricity. After anaerobic digestion, short-chain organic acids are significantly degraded and parasite eggs and some harmful microorganisms are killed. As the main nitrogen is ammonia nitrogen, the digestate is good fertilizer [31]. Therefore, the analysis in depth of the pig slurry composition before planning a treatment technology or using it as a fertilizer has become highly relevant in the current context.

## 5. Conclusions

- Only the concentration of quats in the weaners stage was statistically higher than in the finishers stage ($p \leq 0.05$).
- Pig slurry showed the presence of potentially toxic elements, such as Cu, Zn, and quats.
- The presence of pathogenic microorganisms was detected, such as *E. coli*, *Salmonella* spp., and parasites mainly represented by *Trichuris* spp. and *Trichostrongylus* spp.
- Adverse effects were observed in all the species exposed to pig slurries from concentrations greater than 1.35%.
- Multivariate analyses showed positive correlations between toxicity on *D. magna* and $COD_S$, sulfate, Zn, and Cu, and between phytotoxicity and $COD_T$, ammonium, Na, K, EC, and quats.
- The lowest S/I ratio was associated to high methane yields and removals of $COD_T$, $COD_S$, and VFA, indicating that it improves the efficiency of the anaerobic process.

This study allowed determining the physicochemical properties, microbiological and parasitological content, and ecotoxicological effects of pig slurries and the performance of the BMP test at two different S/I ratios. Future studies including cosubstrate would be necessary to achieve higher methane yields than those found in the present study for pig slurry.

**Supplementary Materials:** The following supporting information can be downloaded at: https://doi.org/10.5281/zenodo.7606317. Figure S1: mean ($\pm$ SD) cumulative methane production for T1 and T2 ($n = 3$). T1: substrate/inoculum ratio of 0.36 g COD/g VSS, T2: substrate/inoculum ratio of 0.62 g COD/g VSS. Figure S2: principal components analysis (PCA) shows the association between physicochemical parameters, removal percentages, biogas production, and treatments used in BMP test. Table S1: characteristics of the BMP test set-up.

**Author Contributions:** Conceptualization, M.E.B.; methodology, M.E.B., B.J.Y. and P.A.B.; formal analysis, B.J.Y.; investigation, M.E.B., B.J.Y., N.I.R. and P.A.B.; writing—original draft preparation, M.E.B.; writing—review and editing, B.J.Y., N.I.R., P.A.B., W.W. and D.K.; funding acquisition, D.E.C.; visualization, M.E.B., B.J.Y. and D.K.. All authors have read and agreed to the published version of the manuscript.

**Funding:** This research was funded by PNNAT 1128042, PD-E2-I518-001, and PE-E7-I149-001 from Instituto Nacional de Tecnología Agropecuaria (INTA), Argentina.

**Institutional Review Board Statement:** Not applicable.

**Informed Consent Statement:** Not applicable.

**Data Availability Statement:** Supporting data reported in this article have been deposited in a publicly archived GitHub repository, accessible at https://github.com/brianjonathanyoung/pig-slurry-properties.git (accessed on 8 February 2023).

**Acknowledgments:** Authors are especially grateful to the Alaralab company for chromatographic analyses.

**Conflicts of Interest:** The authors declare no conflict of interest.

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
