# Peer review of "Relationships among Physicochemical, Microbiological, and Parasitological Parameters, Ecotoxicity, and Biochemical Methane Potential of Pig Slurry"

_sustainability, doi:10.3390/su15043172_

Round 1

Reviewer 1 Report

This manuscript illustrated the relationships among conventional properties, ecotoxicity, pathogens and biogas potential of pig slurry in a typical pig farm from Argentina. In this research, the sample was collected from the pit under weaner stage and finisher stage, which could represent the common quality of pig wastewater in Argentina.

In general, this research provided a lot of data in the field site, including the biogas yield etc., which is the most important parameter for designing the biogas plant and slurry utilization in farmland. For the peer review, I only provide the comments and some suggestion for further research on the biogas fermentation section.

1.     Line 162, what kind of the mixing procedure? How long for one time mixing?

2.     Table 1, pH of the slurry in two stages are a little low, that could be due to a relative long storage time in the pit.

3.     Table 1, there is a large difference of VS/TS between W stage and S stage (58.46% and 71.66%), which would cause different biogas potential.

4.     The BMP test set two S/I ratio (Line 156), but the substrate amounts were different. Therefore, the compare of accumulative biogas production (Line 259, Figure 2 A) was meaningless.

5.     More inoculum leads to high fermentation speed especially in the first several days during the batch fermentation. But Figure 2 B seems opposite result.

6.     Line 411, “whereas observed an association between cumulative methane production and VFA”. This conclusion cannot be researched based on the results in this research.

7.     For further research, the authors could focus on parameters and properties variation before and after biogas fermentation, and hormones, antibiotics, heavy metals and other risk factor in the liquid and solid digestate. What is more, how can reuse the digestate in a safety way which is chief for establishing a planting and breeding cycle.

Author Response

Thank you very much for your positive concluding remarks regarding the manuscript and your constructive comments to help improve it.  English language have been reviewed by a native speaker. Please, find in the attachment the point-by-point replies to all your comments.

Reviewer 2 Report

The manuscript entitled "Relationships among physicochemical, microbiological and parasitological parameters, ecotoxicity, and biochemical methane potential of pig slurry" proposes an interesting subject to ensuring environmental sustenance and promote animal health. The main purpose and approach generally coincided with the scope of the journal. Overall, the main idea of the manuscript is interesting. However, I have comments that authors should address before publication, listed below:

1. In the abstract, it will be ideal to include the figures corresponding to the high contents of Cu, Zn…..etc. mentioned in lines 20-21. Lines 22-23, please indicate the correlation values or probability values here.

Introduction

2.Introduction: This section is sufficiently extensive, it correctly explains the issues of the undertaken research. Much of the cited literature is older than 5 years, and this is very significant in science. I suggest updating selected information. 

3. Line 38. “before applying a waste…” change “a” to “in”

4. The microbiological and parasitological aspect needs to be briefly introduced in this section since they are the main components contributing to the various factors mentioned in the title. What are some of the pathogenic parasite mostly found in these slurries? What studies have currently identified some of them?

Materials and methods

5. The companies and their locations where the chemicals were purchased should be provided.  

6. In sections 2.3 and 2.4, were there any replications to determine the reliability of the tests analysis?

7. Line 104. “according to USEPA” define USEPA on the first mention before subsequently abbreviating.

8. Line 154. What’s the composition of the mineral solution?

Results and discussion

9. Table 1. Why does the Salmonella spp. have no values and indicated as negative and positive?

10. Table 3. Content and values are very difficult to read.

11. Authors should refrain from using “We found” and “In our study” could be “In this study” or use more scientific or technical related terms.

12. Line 293. Any correlation value to indicate?

Conclusion

13. Line 426. “statiscally higher”. The probability value should be indicated.

Author Response

Thank you very much for your positive concluding remarks regarding the manuscript and your constructive comments to help improve it. Please, find in the attachment the point-by-point replies to all your comments.

Reviewer 3 Report

The manuscript “Relationships among physicochemical, microbiological and parasitological parameters, ecotoxicity, and biochemical methane potential of pig slurry” presents interesting results. It is well written and organised. I would like to consider only a few suggestions to improve it in order to be accepted for publication. 

- Table 3, is too small. Please increase font size or reorganised the data preseted in a more suitable way.

- An innoculum characterization with the main physicochemical parameters as the presented for pig slurry samples, might be included.

- How long did Biochemical methane potential tests lenght?

Author Response

(The authors gave the same response as above.)

Round 2

Reviewer 2 Report

The authors have been addressed all comments.